# Rapid Detection of *Escherichia coli* Antibiotic Susceptibility Using Live/Dead Spectrometry for Lytic Agents

**DOI:** 10.3390/microorganisms9050924

**Published:** 2021-04-26

**Authors:** Julia Robertson, Cushla McGoverin, Joni R. White, Frédérique Vanholsbeeck, Simon Swift

**Affiliations:** 1Department of Molecular Medicine and Pathology, The University of Auckland, Auckland 1023, New Zealand; j.white@auckland.ac.nz (J.R.W.); s.swift@auckland.ac.nz (S.S.); 2The Dodd-Walls Centre for Photonic and Quantum Technologies, Auckland 1010, New Zealand; c.mcgoverin@auckland.ac.nz (C.M.); f.vanholsbeeck@auckland.ac.nz (F.V.); 3Department of Physics, The University of Auckland, Auckland 1010, New Zealand

**Keywords:** antibiotic susceptibility, viability, fluorescence, *Escherichia coli*, SYTO 9, propidium iodide

## Abstract

Antibiotic resistance is a serious threat to public health. The empiric use of the wrong antibiotic occurs due to urgency in treatment combined with slow, culture-based diagnostic techniques. Inappropriate antibiotic choice can promote the development of antibiotic resistance. We investigated live/dead spectrometry using a fluorimeter (Optrode) as a rapid alternative to culture-based techniques through application of the LIVE/DEAD^®^ BacLight^TM^ Bacterial Viability Kit. Killing was detected by the Optrode in near real-time when *Escherichia coli* was treated with lytic antibiotics—ampicillin and polymyxin B—and stained with SYTO 9 and/or propidium iodide. Antibiotic concentration, bacterial growth phase, and treatment time used affected the efficacy of this detection method. Quantification methods of the lethal action and inhibitory action of the non-lytic antibiotics, ciprofloxacin and chloramphenicol, respectively, remain to be elucidated.

## 1. Introduction

Antibiotic susceptibility tests (AST) are used in clinical settings to determine an appropriate antibiotic to treat a bacterial infection [1]. The current gold standard ASTs use culture-based methods; however, the usefulness of these methods is constrained by the time taken to generate a result, which is usually at least 48 h [1]. Furthermore, the most commonly used tests for the minimum inhibitory concentration (MIC) measure antibiotic concentrations that inhibit the growth of, but do not necessarily kill, bacteria [2]. A further culture step from antibiotic-inhibited test cultures is needed to determine the minimum bactericidal concentration (MBC) [3]. An increase in the MBC can be observed for bacteria where a sub-population exhibits tolerance or persistence to an antibiotic without a noticeable change in the MIC [4]. As such, an inappropriate antibiotic may be empirically prescribed during this time, leading to potential treatment failure and exposure of bacteria to superfluous antibiotics, which is known to promote the development of antimicrobial resistance (AMR) [5,6]. Rates of antibiotic resistance are increasing worldwide and represent a significant health burden [5].

Improved, faster diagnostics that can inform antibiotic choice in hours rather than days can limit the use of inappropriate antibiotics and the associated problems. A variety of approaches have been proposed, but each has limitations; these include nucleic acid- [1,7,8,9,10,11], structure- [9,10,12,13] (e.g., antibody based), mass spectrometry- [11,14,15,16], fluorescence- [17,18,19,20], imaging- [11,21], and other optical-based methods [22,23,24,25]. No ideal method has yet been developed.

Assessment of antibiotic sensitivity using fluorescent dyes is an attractive alternative to culture-based methods because the dyes allow for near real-time quantification of bacterial viability. Available fluorescent viability dyes report on cell processes that reflect a basic aspect of life [26,27,28], including cellular energy as indicated by membrane potential, an intact cell membrane as indicated by membrane integrity, or metabolism as indicated by enzyme activity [29,30]. Limitations associated with metabolic and membrane potential dyes, including metabolic activity continuing in dead cells [28,31,32] and re-activation of depolarised cells [28,33], led us to select membrane integrity dyes to investigate their potential to establish a rapid AST.

The LIVE/DEAD^®^ BacLight^TM^ Bacterial Viability Kit (BacLight Kit) contains the fluorescent dyes, SYTO 9, and propidium iodide (PI), which can be used to distinguish live and dead cells based on their membrane integrity [34]. Both dyes bind nucleic acid, resulting in an increased fluorescent signal [35]. SYTO 9 is membrane permeable and can enter all cells, resulting in green fluorescence [35,36]. PI is membrane impermeable and can only enter cells with compromised membranes (regarded as dead), resulting in red fluorescence [35,36]. Therefore, live and dead cells can be differentiated based on the relative green and red fluorescence [35,36,37]. Live/dead staining has been investigated for its potential to rapidly determine the proportion of live cells in a population; however, these publications did not try to optimise the protocol beyond the recommendations in the BacLight Kit instructions [38,39,40,41].

We have previously used live/dead spectrometry to determine the viability status of *Escherichia coli* stained with the dyes from the BacLight Kit [42,43]. Suspensions of *E. coli* containing varying ratios of live cells and dead cells generated by treating with 70% isopropanol were stained with SYTO 9 and PI [34,42]. Fluorescence was measured using a fibre-based fluorimeter called the Optrode, which excites the sample and collects the emitted light as a fluorescence spectrum (490–750 nm) [42,43]. The green and red emissions were derived from integrating at 505–515 nm and 600–610 nm, respectively [42,43]. The fluorescence emissions were used to determine the proportion of live cells in a population [42,43]. The BacLight Kit recommends determining the viability status of a population by calculating the ratio of green to red emissions [34]; however, an improved ratio was determined called the adjusted dye ratio (ADR), which indicated the proportion of live cells in a population and generated data with a better fit to viability [42].

The protocol for live/dead spectrometry of *E. coli* was optimised to reflect the desired properties of a rapid AST [42]. In order for a rapid AST to be implemented in a real-world setting, the test needs to have a minimal number of processing steps to improve reliability and reduce costs [1]. To this end, we investigated establishment of a protocol that did not require a sample to be washed before staining, which was achieved by selection of a single medium, minimal A, suitable for both growing and staining the bacteria [42,44]. It was demonstrated that *E. coli* in minimal A generated data with an improved fit compared to saline, which is the recommended staining medium in the BacLight Kit instructions [42].

In our previous work, we observed that SYTO 9 and PI interact differentially when comparing stained live cells and stained dead cells [42]. For the dead cells, both dyes are present in the cells, leading to a quenching/enhancement interaction in which green emissions are reduced with a concomitant increase in red emissions [42]. Therefore, detection of a quenching/enhancement interaction can indicate the presence of dead cells, which can be determined by calculating the difference in green emissions between cells stained with SYTO 9 only and cells stained with SYTO 9 and PI [42]. Examination of the impact of dye interaction on green emissions is an additional measure of viability alongside red fluorescence intensity from PI only stained cells and the ADR. The literature pertaining to application of the BacLight Kit to assess bacterial viability has only used the green:red fluorescence ratio as a measure of the proportion of viable cells [17,39,45,46,47,48]. The work presented here uses the approach outlined in our previous publication [42]**.** Examination of the fluorescence data in more detail (e.g., SYTO 9 only and PI only stained cells) to generate a deeper understanding of live/dead staining of treated cells. We hypothesise that SYTO 9/PI staining and fluorescence spectrum measurements using the Optrode can generate data for rapid viability determination. Our ability to acquire this data offers a novel approach that can be used to evaluate the bactericidal potential of an antibiotic in a near real-time period.

In this work, we applied the optimised live/dead spectrometry assay using the Optrode to detect antibiotic activity against *E. coli* and explored how experimental parameters may impact the results. We selected a range of antibiotics: two antibiotics that target the cell envelope and cause cell lysis, ampicillin and polymyxin B, and two non-lytic antibiotics, ciprofloxacin and chloramphenicol. Ampicillin is a type of penicillin that targets penicillin-binding proteins leading to inhibition of cell-wall synthesis and cell lysis [49]. Polymyxin B is a lipopeptide that targets the outer membrane compromising membrane integrity and causing lysis [50]. Ciprofloxacin is a bactericidal antibiotic that targets DNA synthesis, while chloramphenicol is a bacteriostatic antibiotic that targets protein synthesis [49].

## 2. Materials and Methods

### 2.1. Bacterial Strain

*E. coli* K-12 strain MG1655 (referred to as *E. coli* MG1655) was selected for live/dead detection in this work as it is a well characterised, widely used strain [51]. *E. coli* MG1655 was grown at 37 °C with shaking at 200 rpm where appropriate.

### 2.2. Media and Chemicals

The LIVE/DEAD^®^ BacLight^TM^ Bacterial Viability Kit (L34856) was purchased from Invitrogen. Cell biology reagents were purchased from Sigma-Aldrich. Bacterial growth media and analytical-grade chemicals included sodium chloride, tryptic soy broth (TSB), agar, propan-2-ol (isopropanol), ampicillin sodium salt, polymyxin B sulfate, ciprofloxacin, and chloramphenicol. Antibiotic stock solutions were established fresh for each experiment in water (ampicillin, polymyxin B), ethanol (chloramphenicol), and 0.1 N HCl (ciprofloxacin) at 100×, and the filter was sterilised.

*E. coli* was cultured in a defined medium, minimal A (MM). One litre of a 5× minimal A salts solution was made according to the following amounts: 5 g (NH_4_)_2_SO_4_, 22.5 g KH_2_PO_4_, 52.5 g K_2_HPO_4_, and 2.5 g HOC(COONa) (CH_2_COONa)_2_·2H_2_O (sodium citrate·2H_2_O). After autoclaving, this solution was diluted to 1× with sterile water and the following sterile solutions, per litre: 1 mL of 1 M MgSO_4_.7H_2_O and 10 mL of a 20% *w*/*v* glucose solution.

### 2.3. Culture Preparation

A turbid overnight culture of *E. coli* in MM was habituated in fresh MM at a 1:20 dilution and grown for about 3.5 h at 37 °C with 200 rpm shaking until an optical density at 600 nm (OD_600_) of 0.4–0.6 (exponential phase) was reached. The habituated culture was diluted to ~1 × 10^8^ CFU/mL (an OD_600_ of 0.255) in MM. The inoculum was enumerated by spreading 20 µL of a 10-fold dilution series on tryptic soy agar (TSA) plates with colony forming units (CFUs) counted after a 16 h incubation at 37 °C.

### 2.4. Determination of MIC and MBC

Antibiotic solutions were prepared at 2× the final desired concentration. Exponential phase *E. coli* at ~1 × 10^8^ CFU/mL were challenged with a doubling dilution of these solutions in a 96-well plate. The MIC was defined as the lowest concentration of antibiotic that was able to inhibit growth of test bacteria following a 20 h incubation at 37 °C measured using an absorbance plate reader [2,3]. The wells that had no change in OD_600_ were selected for MBC testing [3]. For this, 10 µL was spread onto 3 TSA plates and CFUs counted after a 16 h incubation at 37 °C. The results of this testing informed the antibiotic concentrations used in the later work (Appendix A).

### 2.5. Antibiotic Challenge

Exponential phase *E. coli* at ~1 × 10^8^ CFU/mL were challenged with 75 µg/mL ampicillin (~1× MBC), 8 µg/mL polymyxin B (1× MBC), 0.25 µg/mL ciprofloxacin (1× MBC), or 16 µg/mL chloramphenicol (1× MIC). We reasoned that, as the LIVE/DEAD^®^ BacLight^TM^ kit measures a parameter of cell viability, the MBC was the most appropriate measure here; however, the MIC of chloramphenicol was used because a sufficiently concentrated stock solution to mediate a 99.9% reduction in viable cells could not be prepared due to limits in solubility. *E. coli* treatments were incubated at 37 °C with 200 rpm shaking. For ampicillin, aliquots were taken at 0 h (before treatment), 0.5 h, 2 h, and 5 h. For polymyxin B, aliquots were taken at 0 h, 0.5 h, and 2 h; pilot data indicated that a 5 h time point was not required due to the rapid action of this antibiotic. For ciprofloxacin and chloramphenicol, aliquots were taken at 0 h, 2 h, and 5 h, reflecting the slower cell-killing action of these antibiotics observed using agar plate counts. Pilot data indicated that a 0.5 h time point was not required for ciprofloxacin and chloramphenicol. Cell viability of the aliquots from antibiotic-treated and untreated cells were determined using live/dead spectrometry (staining with live/dead dyes and measuring the resulting fluorescence using the Optrode) and by culture-based enumeration (enumeration of CFUs from 20 µL aliquots on TSA plates).

Next, 70% isopropanol was used as a control treatment for dead cells as per the LIVE/DEAD^®^ BacLight^TM^ Bacterial Viability Kit instructions [34]. Then, ~1 × 10^8^ CFU/mL exponential phase *E. coli* were harvested by centrifugation at 7000× *g* for 7 min at room temperature and concentrated 10-fold in MM. For live cells, the concentrated culture was diluted 10-fold in MM. For dead cells, the concentrated culture was diluted 10-fold in 70% isopropanol. Live cell and dead cell suspensions were incubated at 28 °C with 200 rpm shaking for 15 min [42] and for 1 h [34]. Before staining, the live and dead cells were harvested at 7000× *g* for 7 min at room temperature and resuspended in fresh MM. Viability of these washed samples was determined using live/dead spectrometry and culture-based enumeration.

For experiments examining the effect of washing a sample before staining, at each time point, duplicate aliquots from treated and untreated cells were taken. One aliquot was left unwashed and the other aliquot was washed, i.e., cells were isolated via centrifugation (7000× *g* for 7 min at room temperature) and resuspended in fresh media. Live/dead staining and measurements were performed on the unwashed and washed samples. The unwashed samples were used for culture-based enumeration. 

For experiments examining the effect of washing a sample after staining, duplicate aliquots were taken and stained with SYTO 9 and/or PI. One aliquot was left unwashed and the other aliquot was washed in fresh media as stated above. Viable cells in the pre-washed samples were enumerated as CFUs on TSA plates.

To determine the impact of antibiotic concentration on live/dead spectrometry, *E. coli* was challenged as before except with ~0.2× MBC and ~2× MBC of ampicillin and ciprofloxacin. To determine the impact of growth phase of live/dead spectrometry, *E. coli* inoculum was prepared in exponential phase as done previously and in stationary phase by growing a 50 mL culture at 37 °C with 200 rpm shaking for 16 h. The exponential phase and stationary phase *E. coli* were challenged with ~1× MBC of ampicillin and ciprofloxacin.

### 2.6. Staining Samples with SYTO 9 And/Or PI

Working solutions of SYTO 9 and PI were established in 0.85% saline in amber microcentrifuge tubes (SSIbio) at 0.0334 mM and 0.4 mM, respectively, and were stored on ice. Stained samples were established by adding 50 µL SYTO 9 working solution, 50 µL PI working solution, and/or 0.85% saline (for a total volume of 100 µL) to 0.9 mL culture in an amber tube. Final concentrations of SYTO 9 and PI were 1.67 µM and 20 µM [42]. For all experiments performed except for the flow cytometry, each aliquot was stained with SYTO 9, PI, and SYTO 9 and PI. For the flow cytometry experiment, aliquots were stained with SYTO 9 and PI only. Following the addition of stain, samples were mixed on a vortex platform for 15 min at room temperature.

### 2.7. Measuring Sample Fluorescence with the Optrode

The Optrode, a fluorometer, was used to excite the samples and then collect and measure the spectrum of the light emitted as a result of the excitation [42]. The Optrode has been described previously [42,52,53]. Measurements were taken using a continuous-measurement script that synchronises the laser and the spectrometer to take 20 ms integration time measurements continuously for a total time of 10 s. Three measurements were taken from each tube and the fibre probe was washed in 70% ethanol between readings. Background readings were taken in triplicate from unstained media. The power of the laser at the sample, kept around 10 mW, was monitored during the measurement using a power meter and photodiode for each experiment.

### 2.8. Processing of Optrode Data Files

All data analyses were carried out using the RStudio (version 1.1.414; R version 3.4.2), MESS [54], and rhdf5 [55] packages. A script in R studio was used to process the data files (hdf5) generated by the Optrode measurements [56]. This script sums the first x collected spectra, normalises the sum to 1 ms and 1 mW, and then subtracts the average background spectrum normalised to the same conditions. The user can define wavelength ranges to be integrated and the number of consecutive spectra to be used for analysis. For the experiments presented in this work, we chose to sum three spectra (effective integration time 60 ms); green emissions were derived from 505–515 nm and red emissions from 600–610 nm [42,53]. The output is a data table containing normalised fluorescence intensities (given as relative fluorescence units, RFU) for the wavelength ranges for each measurement taken.

### 2.9. Flow Cytometry Measurements

Exponential phase *E. coli* was challenged with antibiotics as stated above, except aliquots were taken at all time points for all treatments. Isopropanol-killed cells and untreated live cells were used as controls. Cell viability of the aliquots from treated and untreated cells were determined using live/dead spectrometry, flow cytometry measurements of stained aliquots, and by enumeration as CFUs on TSA plates. Aliquots for flow cytometry were prepared as for the Optrode measurements except for the addition of 10 µL microsphere standard from the LIVE/DEAD^®^ BacLight^TM^ Bacterial Viability Kit.

Flow cytometry measurements were taken on an LSR-II flow cytometer (Becton Dickinson) with the threshold set to side scatter (SSC) and flow rate set to the lowest possible, 6 µL/minute [57]. All measurements ran for 10,000 events to ensure a statistical coefficient of variation (CV) less than 1% [58]. Parameters were used as follows: excitation by a 20 mW, 488 nm laser; 505 nm longpass filter and a 530/30 nm bandpass filter for SYTO 9; and a 685 nm longpass filter and 695/40 nm bandpass filter for PI. The number of microsphere beads added to each sample was used to calculate the absolute concentration of live and dead bacteria measured, via the bead-based FCM method [52].

### 2.10. Biological Data Processing

Previous work allowed for a set of criteria to be determined that indicate a cell has a damaged membrane and is thus presumed to be dead [42]: (1) The difference in green emissions from SYTO 9-only stained samples and from SYTO 9- & PI-stained samples (normalised to SYTO 9 emissions) is approaching 0.91, (2) red emissions from PI-only stained cells is greater than or equal to 4 × 10^3^ RFU, and (3) adjusted dye ratio (ADR) is approaching 28.

For the first criteria, we have previously demonstrated that SYTO 9 and PI interact when bound to nucleic acid in cells with compromised membranes [42]. The nature of this interaction is quenching of the SYTO 9 signal with concomitant enhancement of the PI signal [42]. In contrast, for cells with intact membranes, a cross-talk interaction occurs in which the emissions from SYTO 9 overlap the wavelength emission range from PI [42]. Both types of dye interaction result in an increased PI signal, while only the quenching/enhancement interaction impacts the SYTO 9 signal. Therefore, the presence of cells with damaged membranes can be indicated by diminished green emissions from SYTO 9- and PI-stained cells relative to SYTO 9-only-stained cells.

The fluorescence intensities of SYTO 9 only and SYTO 9 and PI derived from integration at a defined wavelength range can be used to determine the occurrence of the quenching/enhancement interaction, defined in Equation (1) [42].
(1)$$\mathrm{Normalised}\;\mathrm{difference}\;\mathrm{in}\;\mathrm{green}\;\mathrm{emissions}=\frac{\left(\mathrm{SYTO}\;9-\mathrm{SYTO}\;9\;\&\;\mathrm{PI}\right)}{\mathrm{SYTO}\;9}$$
where SYTO 9 and SYTO 9 and PI are the fluorescence intensities integrated at 505–515 nm.

The reference value of 0.91 corresponds to 100% dead cells in the measured population and was derived from the median of triplicate measurements of isopropanol killed cells—the positive control recommended in the BacLight Kit instructions [34].

For the second criteria, the red emissions derived from integration at 600–610 nm of PI-only-stained cells were used. The reference value of 4 × 10^3^ RFU was derived from the median of triplicate measurements of isopropanol-killed cells. 

For the third criteria, the fluorescence intensities derived from integration at 505–515 nm and 600–610 nm from SYTO 9- and PI-stained cells can be used to generate the ADR—the proportion of live cells in a sample—defined in equation (2) [42,53].
(2)$$\mathrm{ADR}=\mathrm{proportion}\;\mathrm{of}\;\mathrm{live}\;\mathrm{cells}\propto \left(100\times \frac{\mathrm{SYTO}\;9}{\mathrm{PI}}\right)/\left(1+\frac{\mathrm{SYTO}\;9}{\mathrm{PI}}\right)$$
where SYTO 9 and PI are the fluorescence intensities integrated at 505–515 nm and 600–610 nm for SYTO 9 and PI, respectively. The ADR reference value of 28 was derived from the median of triplicate measurements of isopropanol-killed cells.

### 2.11. Processing of Flow Cytometry Data

All data was exported from FACSDiva (Becton Dickinson) as fcs files and imported into FlowJo V10 (FlowJo, LLC) for analysis [57]. Software compensation was applied to all measurements based on measurements from SYTO 9-stained and PI-stained live cells [57]. For ampicillin-treated cells, compensation was SYTO 9 = −2 and PI = 2, while for all other samples, compensation was SYTO 9 = 6 and PI = 0.

Gating was manual, and each technical replicate for a single sample had the exact same gates applied [57]. Noise, including cellular debris, growth media, and electronic noise, was excluded from gates. Four subpopulations were discerned based on their staining characteristics, reflecting their different physiological states. These subpopulations were: (1) lower left section being unstained debris, (2) upper left section being PI-labelled dead cells, (3) lower right section being SYTO 9-labelled live cells, and (4) upper right section being injured/dying cells that are labelled with both SYTO 9 and PI [59].

### 2.12. Statistical Analysis

To determine the impact of washing samples before or after staining, a two-way repeated measurement analysis of variance (two-way RM ANOVA) with Tukey multiple comparisons test was used to compare washed and unwashed samples with a *p* value of less than 0.05 considered significant.

For the antibiotic challenge experiments, a two-way RM ANOVA with Sidak’s multiple comparisons test was used to determine statistical significance for each treatment and challenge time combination relative to the untreated control with a *p* value of less than 0.05 considered significant. To determine the impact of antibiotic concentration or growth phase on live/dead spectrometry, the difference between treated and untreated was analysed using a two-way RM ANOVA with Tukey multiple comparisons test with a *p* value of less than 0.05 considered significant.

To determine the similarity between the live/dead spectrometry measurements from the original antibiotic challenge experiments and the flow cytometry experiments, a two-way RM ANOVA with Tukey multiple comparisons test was used with a *p* value of less than 0.05 considered significant.

## 3. Results

We investigated use of live/dead spectrometry to detect antibiotic-mediated killing of *E. coli* in near real-time. Model lytic and non-lytic antibiotics, ampicillin and ciprofloxacin, respectively, were used to challenge *E. coli*. Initial optimisation work focused on the impact of washing the samples before and after staining with SYTO 9 and/or PI. Following this, the optimised protocol was used to determine viability of an *E. coli* population challenged with a range of antibiotics using live/dead spectrometry. The results from this work were validated using flow cytometry. Finally, the protocol was further optimised through examination of the impact of antibiotic concentration, bacterial growth phase, and treatment time on the live/dead spectrometry of cell populations challenged with the model antibiotics.

### 3.1. The Effect of Washing Samples before and after Staining

We want to establish an accurate and technically facile AST. To facilitate this, we investigated the impact of washing cells after antibiotic challenge and after live/dead staining upon the final fluorescence signal. We hypothesised that washing cells would cause the loss of biological information from lysed cells, leading to a reduction in both total fluorescence and the ratio of green and red fluorescence emissions from which the ADR is derived.

We challenged *E. coli* with model lytic (ampicillin) and non-lytic (ciprofloxacin) antibiotics and duplicate aliquots were taken from each treatment. Cells in one aliquot were left unwashed and those in the other were washed in fresh media before live/dead spectrometry was performed. For *E. coli* challenged with ampicillin and stained with SYTO 9 or PI, statistically significant lower green and red fluorescence intensities were observed for washed samples compared to unwashed samples (Figure 1A,C) (*p* value < 0.05). The difference between unwashed and washed samples increased over time for SYTO 9-stained ampicillin-challenged cells (Figure 1A). The ADR, which represents the proportion of live cells in a SYTO 9- and PI-stained sample, was lower for unwashed samples relative to washed samples. The higher ADR of the washed samples reflects the reduced green emission compared to red.

For *E. coli* challenged with ciprofloxacin, the washed sample differed significantly from the unwashed sample for the impact of dye interaction on green emissions (Figure 1B) (*p* value < 0.05). The significance likely reflects the difference at the pre-treatment measurement. Otherwise, for untreated and ciprofloxacin treated *E. coli*, no significant difference was found for fluorescence intensities between unwashed and washed samples (Figure 1B,D,F). The results presented in Figure 1 support the hypothesis of using unwashed samples of antibiotic-challenged cells for live/dead spectrometry.

Washing stained cells in samples before live/dead spectrometry samples can be performed to remove the unbound dye that contributes to background fluorescence. The results presented in Figure 1 led us to hypothesise that washing samples containing lysed cells post-staining would be detrimental to the detection of killing by ampicillin treatment; however, it had little impact on the detection of killing by ciprofloxacin or by isopropanol in controls.

For isopropanol-challenged cells, washing unbound SYTO 9, unbound PI, and unbound SYTO 9 and PI resulted in decreased green emissions, decreased red emissions, and higher ADR, respectively (Figure 2). For the untreated control, similar trends were observed for PI-stained and SYTO 9- and PI-stained samples. These differences in intensities between unwashed and washed samples were statistically significant (*p* value < 0.05).

For ampicillin-treated cells, washing unbound SYTO 9 from samples resulted in significantly decreased green emissions (Figure 3A) compared to unwashed cells (*p* value: <0.05). For all treatments, washing unbound PI from samples resulted in significantly decreased red emissions and higher ADR values (Figure 3C,F) compared to unwashed cells (*p* value < 0.05). The decrease in red emissions from all washed cells reflects removal of unbound dye that generates a background signal. For ampicillin-challenged cells, the reduced green and red emissions are also derived from washing-mediated loss of DNA from lysed cells, which is supported by the difference between emissions from washed and unwashed samples increasing concurrently with the number of lysed cells.

The benefit of the lower background signal facilitated by washing unbound PI from the samples—increased sensitivity of bound PI detection—is outweighed by the disadvantage of reduced fluorescence intensities derived from loss of biological information in ampicillin-challenged samples resulting in decreased sensitivity of detection of the loss of viability through cell lysis. Therefore, from this point, antibiotic killing assays were performed using unwashed samples.

### 3.2. Live/Dead Spectrometry for Viability Determination of Unwashed Samples of Antibiotic-Challenged E. coli

*E. coli* was challenged with ampicillin and polymyxin B (lytic antibiotics), ciprofloxacin (non-lytic antibiotic), and chloramphenicol (non-lytic antibiotic). We hypothesised that loss of *E. coli* viability resulting from challenge with lytic antibiotics can be detected using live/dead spectrometry, reflecting the targeting of the membrane acting to increase access of the dyes to intracellular nucleic acid. We hypothesised that the loss of *E. coli* viability due to challenge with non-lytic antibiotics that have non-membrane cellular targets would be detectable at later time points using live/dead spectrometry.

For *E. coli* challenged with ampicillin, the three dead cell criteria for live/dead spectrometry all indicate that the cells are dying over time (Figure 4A,C,E). The impact of dye interaction on green emissions suggests that cell death occurs from 0.5 h exposure time, resulting in most of the population dead by 5 h (Figure 4A). Ampicillin-treated cells were significantly different compared to untreated cells for all time points (*p* value < 0.05). Red emissions from ampicillin-treated cells stained with PI increased to the level measured for a population of isopropanol-killed cells at the 0.5 h exposure time (Figure 4C), which was significantly different from untreated cells at 0.5 h (*p* value < 0.05, Sidak’s multiple comparisons test). The ADR also suggests that cell death occurs from 0.5 h time point, increasing with exposure time (Figure 4E). The ADR of ampicillin-treated cells was significantly different from the ADR of untreated cells at 0.5 h–5 h exposure times points (*p* value < 0.05). Culture-based enumeration indicated that a minimum of a 2 h treatment time is needed to mediate noticeable cell death (Figure 4E). Plate counts of treated cells were significantly different from those of untreated cells at 2 h and 5 h exposure times (*p* value < 0.05). Thus, live/dead spectrometry after 0.5 h treatment can predict the effectiveness of ampicillin in culture-based assays.

For *E. coli* challenged with polymyxin B, the three dead cell criteria for live/dead spectrometry all indicate that the majority, if not all, of the cell population was dead following a 0.5 h treatment (Figure 4B,D,F). For the impact of dye interaction on green emissions, the polymyxin B-challenged cells were significantly different from untreated cells at both challenge times. However, for red emissions, this was only the case for the 0.5 h treatment time (*p* value < 0.05). Culture-based enumeration suggests that significant reduction in viability occurred following the 0.5 h treatment; however, a 2 h exposure time is needed to kill all the cells (Figure 4F). Plate counts of polymyxin B-treated cells were significantly different from those of untreated cells at 0.5 h and 2 h exposure times (*p* value < 0.05). Thus, live/dead spectrometry after 0.5 h treatment can predict the effectiveness of polymyxin B in culture-based assays. 

For *E. coli* challenged with ciprofloxacin, the impact of dye interaction on green emissions suggests that cell death (as defined by the BacLight Kit) has not occurred; rather, as treatment time increases, the treated cells trend away from the dead population value of 0.91 (Figure 5A). The red fluorescence intensities from treated cells stained with PI is the same as for untreated cells, implying that no cells have died (Figure 5C). The ADR of ciprofloxacin-treated cells remains stable over time, suggesting no change in the viability status of the treated cells, while the ADR of untreated cells increases at 2 h and returns to the initial level by 5 h (Figure 5E). The difference between the ADR of treated and untreated cells is statistically significant (*p* value < 0.05,); however, this difference is attributed to the growth of the untreated cells rather than due to the impact of ciprofloxacin on *E. coli* viability. Culture-based enumeration of ciprofloxacin-treated cells indicated that significant killing of *E. coli* occurred from 2 h (Figure 5E; *p* value < 0.05). Thus, live/dead spectrometry was unable to predict the lethal activity of ciprofloxacin seen in culture-based assays.

For *E. coli* challenged with chloramphenicol, two of the three dead cell criteria suggest that killing had not occurred. The impact of dye interaction on green emissions of chloramphenicol-challenged cells trended away from the dead cell population value (Figure 5B); however, the differences were not significant. Challenge with chloramphenicol produced similar results to ciprofloxacin for the impact of dye interaction on green emissions and the red fluorescence intensities, implying that no cells had lost membrane integrity (Figure 5D). The ADR for chloramphenicol-challenged cells, however, differed from ciprofloxacin, as the former decreased by 2 h (Figure 5F), implying that some cells in the population had died. The difference between the ADR of treated and untreated cells is statistically significant (*p* value < 0.05). Culture-based enumeration showed that chloramphenicol inhibited the growth of treated cells (Figure 5F). The difference between the plate counts of treated and untreated cells is statistically significant (*p* value < 0.05); however, this is attributed to the growth of the untreated cells rather than due to the impact of chloramphenicol on *E. coli* viability. Thus, while both live/dead spectrometry and culture-based methods were unable to detect cell killing by chloramphenicol at 1× MIC in the 5 h assay period, there is evidence of growth inhibition when comparing ADR values between treated and untreated cells.

The results presented demonstrate that loss of viability from lytic antibiotics can be detected using live/dead spectrometry (Figure 4); however, the assay was not sensitive to the lethal action of the non-lytic antibiotic, ciprofloxacin (Figure 5). Following this, the results generated from the Optrode-based experiments were verified using flow cytometry.

### 3.3. Validation of Live/Dead Spectrometry for Viability Determination of Unwashed Samples of Antibiotic-Challenged E. coli

To validate the results obtained with the Optrode, the viability of antibiotic-challenged *E. coli* was determined using flow cytometry after live/dead staining. We measured antibiotic-treated and untreated samples stained with SYTO 9 and PI at 0.5 h, 2 h, and 5 h exposure times on the Optrode and an LSR-II flow cytometer (BD). The viable cell count was also determined using culture-based enumeration. Appendix A present a comparison of the Optrode data from the original experiments (Figure 4 and Figure 5) and the FCM experiments (Figure 6 and Figure 7). No statistically significant differences between the measures of viability were found between the original experiments and the FCM experiments (*p* value < 0.05).

Flow cytometry measurements allow determination of the total cell count, live cell count, dying cell count, and dead cell count. The limit of quantification (LoQ) was defined as the lowest concentration of cells that could be detected reliably using this assay, which was 1 × 10^5^ cells/mL. For untreated cells, dead count was mostly below the LoQ, while the total and FCM-live counts were ~10^8^ cells/mL (Figure 6A,E). For isopropanol-challenged cells, the dead count at ~10^8^ cells/mL was representative of the total cell count (Figure 6C). For ampicillin-challenged cells, the 0.5 h time point suggested that the population contained a mix of live, dying, and dead cells (Figure 7A). As time progressed, the total, live, and dying cell counts decreased while the dead count increased (Figure 7A). For polymyxin B-challenged cells, the 0.5 h time point suggested that there was a small population of dying cells while the rest of the population were dead (Figure 7C). The later time points show that the whole population was dead, and the total cell count decreased (Figure 7C). For ciprofloxacin- and chloramphenicol-challenged cells, the total and FCM-live counts did not change over time while the dying cell count increased, with a greater increase observed for the former (Figure 7E,G).

Fluorescence-based viability determination between the two techniques was compared using the ADR for Optrode measurements and live cell counts for flow cytometry measurements. For untreated cells, the ADR and live cell counts agreed, except for the 5 h time point at which the ADR decreased (Figure 6B). For isopropanol-challenged cells, the measurements of viability all indicated that the population was dead (Figure 6D). For ampicillin treated cells, the 0.5 h time point showed some discordance between the ADR and FCM-live counts, with the former indicating lower population viability than the latter (Figure 7B). The measurements demonstrated similar viability levels for the 2 h challenge time; however, the FCM-live counts had reached the LoQ, which meant that viability of the 5 h challenge time was not accurately reported on using this measure (Figure 7B). For polymyxin B-challenged cells, the viability measurements indicated that the total population was dead by 0.5 h (Figure 7D). For ciprofloxacin-challenged cells, ADR and FCM-live counts both indicated that there had been no loss of viability, which contrasts with the culture-based determination of viability (Figure 7F). For chloramphenicol-challenged cells, the FCM-live counts were constant over time, while the ADR suggested some effect on viability (Figure 7H).

In general, FCM supported Optrode measurements for determining the viability of the population. There were some discrepancies between the measurements for untreated cells at 5 h, ampicillin-treated cells at 0.5 h, and chloramphenicol-treated cells at 2 h.

### 3.4. Effect of Antibiotic Concentration on Live/Dead Spectrometry of Challenged E. coli

Following the encouraging outcome of successful detection of killing of *E. coli* by the lytic antibiotics, ampicillin and polymyxin B, using live/dead spectrometry, the assay was further optimised to explore how experimental parameters may impact viability detection. The experimental parameters investigated include antibiotic concentration, bacterial growth phase, and treatment time. Ampicillin and ciprofloxacin were selected for this work as the activity of the former can be detected in our assay and the activity of the latter cannot be detected, providing an interesting comparison.

For unknown clinical samples, the MBC of a strain will not be established, which could result in use of a too high or low antibiotic concentration in an AST. Since the bacterial concentration relative to the antibiotic concentration can influence the lethal outcome of treatment—termed the inoculum effect—we explored the impact of challenging with a high and a low antibiotic concentration on live/dead detection [60]. *E. coli* was challenged with ~0.2× MBC and ~2× MBC of ampicillin or ciprofloxacin.

For *E. coli* challenged with ~0.2× MBC ampicillin, live/dead spectrometry suggested that significant cell death occurred at 2 h and 5 h treatment (Figure 8A,C,E; *p* value < 0.05); however, plate counts showed that no reduction in the number of viable cells remaining had occurred (Figure 8G). For the green emissions, the treated cells were significantly different from the untreated cells at 0.5 h; however, this is due to growth of the untreated cells (Figure 8A). Therefore, use of a sub-lethal antibiotic concentration can generate live/dead spectrometric data that indicates *E. coli* has died when culture methods show that the cells are still viable.

For *E. coli* challenged with ~2× MBC ampicillin, live/dead spectrometry indicated that cell death occurred from 0.5 h exposure time (Figure 8A,C,E). The difference between treated and untreated cells at every time point was statistically significant except for the 2 h and 5 h time points for red emissions (*p* value < 0.05). Culture-based enumeration suggests that most of the population was still viable at the 0.5 h treatment time and required at least a 2 h exposure to kill the cells (Figure 8G). Therefore, these results demonstrate that reliable detection of cell death is dependent on both antibiotic concentration and treatment time.

For *E. coli* challenged with both ~0.2× MBC and 2× MBC ciprofloxacin, live/dead spectrometry did not detect cell death (Figure 8B,D,F). However, a difference in ADR between treated and untreated samples suggests that this method detected the absence of cell growth (Figure 8F). For the impact of dye on green emissions, 0.2× MBC ciprofloxacin treatment at 0.5 h and 2× MBC ciprofloxacin treatment at 2 h and 5 h differed significantly from untreated cells (*p* value < 0.05); however, the treated cells trended away from the dead population value of 0.91. The difference between the ADR for both treatments and untreated (Figure 8F) was statistically significant (*p* value < 0.05), which is due to growth of untreated cells. In contrast to the live/dead spectrometry, culture-based enumeration demonstrated that the number of viable *E. coli* cells remaining was reduced by 3 log and 5 log CFU/mL after a 5 h challenge with 0.2× MBC and 2× MBC ciprofloxacin, respectively (Figure 8H). For culture-based enumeration, 0.2× MBC ciprofloxacin treatment at 2 h and 5 h, and 2× MBC ciprofloxacin treatment at all time points differed significantly from untreated (*p* value < 0.05). Thus, the results presented demonstrate that loss of *E. coli* viability from ciprofloxacin action cannot be detected using live/dead spectrometry, supporting earlier results.

Spectrometry data of *E. coli* challenged with a low concentration of ampicillin detected cell death of most of the population by 5 h (Figure 8). However, the culture-based enumeration showed no reduction in viable cell numbers (Figure 8G). For *E. coli* challenged with a high concentration of ampicillin, reliable detection of loss of viability only occurred at the 2 h and 5 h time points. Therefore, these results demonstrate that both treatment time and antibiotic concentration are important experimental parameters to optimise to reliably detect loss of viability of cells challenged with a lytic antibiotic.

### 3.5. Effect of Bacterial Growth Phase on Live/Dead Spectrometry of E. coli

To further investigate experimental parameters that may impact viability determination of challenged *E. coli*, the effect of bacterial growth phase was examined. It is recommended in the BacLight Kit instructions to stain *E. coli* in its exponential phase as that growth phase generates data that has a good correlation to culture-based assays [34]. For unknown clinical samples, the bacteria may be in a different growth phase, including stationary phase. It is known that the environment and the resulting physiological status of the cells can influence viability staining [35,37,61]. Therefore, we investigated the effect of growth phase on live/dead spectrometry of *E. coli* challenged with ~1× MBC of ampicillin and ciprofloxacin.

The impact that growth phase has on sensitivity to antibiotics (Figure 9) is demonstrated by the significant difference found between green and red emissions from exponential- and stationary-phase cells challenged with ampicillin for 0.5 h and on ADR for all challenge times (*p* value < 0.05). For untreated cells, the different response of exponential- and stationary-phase cultures is shown at 0.5 h time point for ADR (*p* value < 0.05); at later time points, the exponential-phase culture would have reached stationary phase and thus no significant difference was found.

For ampicillin-challenged cells, the impact of dye interaction on green emissions and red emission intensities indicated that most of exponential and stationary cells were killed by 0.5 h and 2 h, respectively (Figure 9G). The differences between exponential-phase, ampicillin-treated and untreated cells for these measurements at 0.5 h to 5 h were statistically significant (*p* value < 0.05). The differences between stationary-phase, ampicillin-treated and untreated cells for these measurements at 2 h and 5 h were statistically significant (*p* value < 0.05).

Overall, live/dead spectrometry of both exponential- and stationary-phase, ampicillin-challenged cells suggested a reduction in viable cells after 2 h and 5 h exposure times, which generally agrees with culture-based enumeration. However, the live/dead spectrometry suggested greater reductions in viability for stationary-phase cells than indicated by the culture-based enumeration. The impact of dye interaction on green emissions and the red fluorescence intensities suggested that most or all of the population was killed, respectively, which is not supported by the plate counts (Figure 9A,C,G). The trends for the ADR and culture-based enumeration of exponential- and stationary-phase cells were similar. From this, it can be concluded that general loss of viability of stationary-phase cells can be detected using live/dead spectrometry; however, this will not be as accurate an indication of culture-based enumeration as for exponential-phase cells.

For ciprofloxacin-challenged cells, the live/dead spectrometry for exponential- and stationary-phase cells were similar (Figure 9B,D,F) except for the impact of dye interaction on green emissions of stationary-phase cells at 5 h, for which these cells differed significantly from the corresponding untreated cells (*p* value < 0.05). At this time point, the treated cells trended away from the dead population value of 0.91 (Figure 9B).

As found previously, the ADR of ciprofloxacin-treated cells remains stable over time, suggesting no change in the viability status of the treated cells, while the ADR of untreated cells increases at 2 h and returns to the initial level by 5 h (Figure 9F). For the ADR, treated cells differed significantly from untreated cells at 0.5 h and 2 h for exponential- and stationary-phase cells (*p* value < 0.05). Culture-based enumeration indicated that exponential-phase cells were reduced by ~1 log more than for stationary-phase cells (Figure 9H). The difference between exponential- and stationary-phase treated cells and the corresponding untreated cells at 2 h and 5 h was statistically significant (*p* value < 0.05). Loss of viability of ciprofloxacin-treated cells could not be detected using live/dead spectrometry and the growth phase of challenged *E. coli* did not have a meaningful impact on this.

The investigation of experimental parameters important for reliable live/dead spectrometry of antibiotic-challenged cells indicated that antibiotic concentration, bacterial growth phase, and treatment time require consideration during assay optimisation.

## 4. Discussion

Live/dead staining offers an attractive technique to rapidly evaluate antibiotic sensitivity with the adjusted dye ratio (ADR; Equation (2) [42]), providing a useful measure of the lethality of an antibiotic. The most commonly used antibiotic sensitivity tests (ASTs) rely on the determination of an antibiotic’s ability to inhibit bacterial growth in measurement of the MIC. At least a further day is required to determine the MBC. If the ratio of MBC:MIC is greater than 4, the antibiotic is termed bacteriostatic; if the ratio is less than or equal to 4, the antibiotic is bactericidal [62]. Two systematic reviews have challenged the dogma that bactericidal antibiotics perform better clinically [63,64]; however, it is stressed that bacteriostatic antibiotics are still able to be lethal to bacteria at a high enough concentration [64]. In this study, we have investigated the suitability of live/dead spectrometry as a basis for a rapid AST for lytic and non-lytic antibiotics.

Recommended protocols for live/dead staining remove media and unbound dye before taking fluorescence measurements [34]. However, for this type of analysis, we recommend omitting sample washing steps. The retention of stained nucleic acid from lysed cells improved detection of bacterial killing. We chose minimal A media for this study as it does not contain components that contribute to the background fluorescence; this is a compromise, as bacteria become less resistant to antibiotics when growing rapidly in complex media (e.g., Mueller Hinton Broth or Tryptic Soy Broth) that contribute a high fluorescent background when compared to the slower rate of growth seen in minimal media [16]. Finally, the simplification of the assay protocol outweighed the benefit of increased sensitivity to detect PI staining in cells challenged with a non-lytic antibiotic. Removing unbound dye should increase fluorescence assay sensitivity by minimising background fluorescence [42,43].The quantum yield of a fluorophore is defined as the ratio of photons emitted to those absorbed, and, therefore, it influences the intensity of light released [65]. The quantum yields of unbound SYTO 9 and PI are low (less than 0.01); however, when bound to nucleic acid, the quantum yield for SYTO 9 increases greatly (to 0.4), while the increase for PI is modest (to 0.12) [66,67,68]. Therefore, washing unbound PI from a sample will mediate a greater decrease in background signal than for SYTO 9, particularly for samples of cells with intact membranes that have high levels of unbound PI. As an example, for ciprofloxacin-challenged cells, removing unbound PI revealed that these cells have higher red fluorescence intensities than untreated cells (Figure 3D), which is not apparent in the unwashed samples. The ADRs of ciprofloxacin-challenged cells and untreated cells were higher for washed samples; however, this did not impact viability determination. 

Our optimised rapid AST performed well at detecting loss of *E. coli* viability sustained from a 2 h challenge with the lytic antibiotics, ampicillin and polymyxin B. The impact of challenge with the non-lytic antibiotics, ciprofloxacin and chloramphenicol, could not be quantified using this assay; however, differences in the live/dead spectrometric data between treated and untreated were observed, which suggests growth inhibition.

It is established that antibiotic concentration, bacterial growth phase, and treatment time can all affect results of AST [60,69,70]. We investigated the impact of these experimental parameters on live/dead spectrometry for rapid AST. Reducing the concentration of ampicillin (0.2× MBC) affected fluorescence-based viability determinations with a false indication of lethal activity (Figure 8). We propose that the increased fluorescence of these cells reflects generation of holes in the cell membrane by ampicillin, which the cells can repair and retain their viability. A similar situation is apparent for cells challenged with 1× MBC of ampicillin for 0.5 h (Figure 4). The increase in emissions with the absence of cell death as indicated by culture would be due to dying cells (as indicated in the FCM data; Figure 7A) which are characterised by ampicillin-mediated increases in membrane permeability that is not yet sufficient to cause loss of viability.

The molecular content of bacteria changes with growth phase reflected in changes in gene expression [71] and protein content [72] that can influence staining with fluorescent dyes. Overlain on this is the greater sensitivity of actively growing cells to antibiotics [16]. Actively growing cells can have transient increases in membrane permeability [35], and nutrient scarcity in stationary phase can cause cell-wall alterations reducing dye permeability [37]. Growth phase influenced the spectrometry results for ampicillin-treated cells (Figure 9); although the overall trend observed describing antibiotic sensitivity was valid. Challenge time was identified as an important parameter in several experiments (Figure 4, Figure 8 and Figure 9). Therefore, to establish a rapid AST using live/dead spectrometry, the protocol would need to be optimised in terms of antibiotic concentration, bacterial growth phase, and challenge time, which would be dependent on both the species and antibiotic [73]. Overall, these results suggest that live/dead spectrometry can be best used to determine lytic antibiotic-mediated *E. coli* death at a 2 h challenge time.

Our rapid AST assay could not quantify cell death caused by exposure to the non-lytic antibiotic, ciprofloxacin. Our results from FCM demonstrated that the dying cell population increases with challenge time (Figure 7E). Despite a portion of the ciprofloxacin treated *E. coli* population staining with PI, this does not reliably indicate viability indicated by a lack of growth on media [74,75,76], reflecting the notion that ciprofloxacin does not have a direct action on the membrane [75,77]. We hypothesise that there is a population of cells with membranes that have lower integrity than other cells; however, they were not detected using the Optrode, as the emissions were eclipsed by those from the rest of the population.

For chloramphenicol-challenged cells, green emissions decreased while red emissions remained at pre-treatment levels, resulting in a decrease in the ADR. The unchanged red emissions indicate that chloramphenicol does not cause a change in membrane permeability [78]. The decrease in green emissions could be due to condensing of the nucleoid in response to chloramphenicol [79,80]; however, this needs further investigation. Distinction between the decrease in the ADR from inhibition compared to lethal action remains to be elucidated. The impact on green emissions from SYTO 9-only stained cells demonstrates the usefulness of live/dead determination using the Optrode, which can easily measure emissions from cells stained with one or more dyes. FCM typically involves measuring emissions of cells within samples stained with two or more dyes, meaning that the impact of dye interaction on the intensity of green emissions cannot be determined [81].

Our results have highlighted an important consideration for using fluorescent viability dyes to detect antibiotic action. The success in determining loss of viability is dependent on the cell process disrupted by the treatment being the same as the cell process reported on by the dye, which is illustrated by the success in detecting the action of lytic antibiotics using membrane-integrity dyes. The limitation of needing to match the dye to the antimicrobial mechanism in a rapid AST may be overcome through application of a range of fluorophores or through development of new viability dyes [82,83]. The work presented in this article also demonstrates that application of SYTO 9 and PI can give insight into the mechanism of action of an antimicrobial treatment when used both in combination and separately, which has potential to inform future investigations on development of a rapid AST. However, there are limitations, including the inability to distinguish and quantify non-lytic antibiotic action. These shortcomings may be overcome by improving data analysis, measurement strategies, and/or the fluorescent stains used. We are investigating using more sophisticated, multivariate analytical methods to probe the data for spectral signatures to indicate mechanism-specific cell death as well as examining the potential to measure emissions from single cells using the Optrode.

Live/dead spectrometry of antibiotic-challenged cells may be enhanced by measuring emissions from single cells [84]. Axenic bacterial cultures are heterogeneous in nature and in response to an antimicrobial challenge, will give rise to physiologically distinct populations [75]. One or more of these populations may go undetected in a population measurement if present in low enough numbers, yet could still have an impact on the outcome of treatment [4,85]. Measurement of emissions from single cells can be achieved using flow cytometry and using microfluidic devices. FCM of samples from the antibiotic challenge assay agreed with Optrode measurements (Figure 6 and Figure 7) except for ampicillin-challenged cells at 5 h. The proportion of FCM-live counts increased from 2 h to 5 h, while the ADR decreased. This is likely due to the way FCM analysis quantifies cell number; FCM cannot detect lysed cells, causing the total cell count to decrease and the relative live-cell concentration to increase. In comparison, the Optrode measures a population, which includes nucleic acids from whole and lysed cells. Therefore, the Optrode can better detect loss of viability from lytic antibiotics.

Microfluidic devices can be used to isolate, concentrate, and detect the response of individual bacterial cells to antibiotic treatment [83]. Microfluidic-based approaches use less resources and thus have reduced running costs, require only small sample sizes, are less laborious, have potential to be automated, have high sensitivity, and are portable [84,86]. Single cell measurements would remove the impact of background signal and increase sensitivity, which would be most beneficial for measuring activity of non-lytic antibiotics (Figure 3D). Measurements taken with a microfluidic device would have a lower limit of detection than for flow cytometry measurements [43,87,88]. The limitation of lysed cells being missed in single cell measurements could be overcome for Optrode measurements by performing a population measurement on the discarded fluid or comparing pre- and post-treatment cell counts.

An ideal rapid AST would: be accurate, cost effective, generate results in 1 h, require a feasible sample size, be portable, have a simple operation, isolate the pathogen from polymicrobial samples, be able to measure unprocessed samples, and have potential for multiplexed and/or high-throughput measurements [1,84,89,90]. Our rapid AST using the Optrode fulfils some of these criteria—portability, simple operation, cost effectiveness, and potential for multiplexed measurements. However, there are limitations restricting its application clinically, including the requirement for a high inoculum, indirect sampling and the need for an isolation/enrichment step, and the time-to-result, which while it is shorter than the standard culture-based methods, is not sufficiently short for a rapid AST. Improvements through the development of microfluidic-based measurements may help with these issues.

## 5. Conclusions

The non-washing protocol for live/dead spectrometry using the Optrode can quantify the lethality of lytic antibiotic action on *E. coli*, provided the protocol is optimised for challenge time (2 h), antibiotic concentration (1× MBC), and bacterial growth phase (ideally exponential). The inhibitory and lethal action of non-lytic antibiotics could not be quantified; however, application of more sophisticated analysis or measurement techniques may improve detection. The live/dead spectrometry assay using the Optrode fulfils some of the attributes of an ideal rapid AST, although more work is required before this AST is usable in clinics.

## Figures and Tables

**Figure 1 microorganisms-09-00924-f001:**
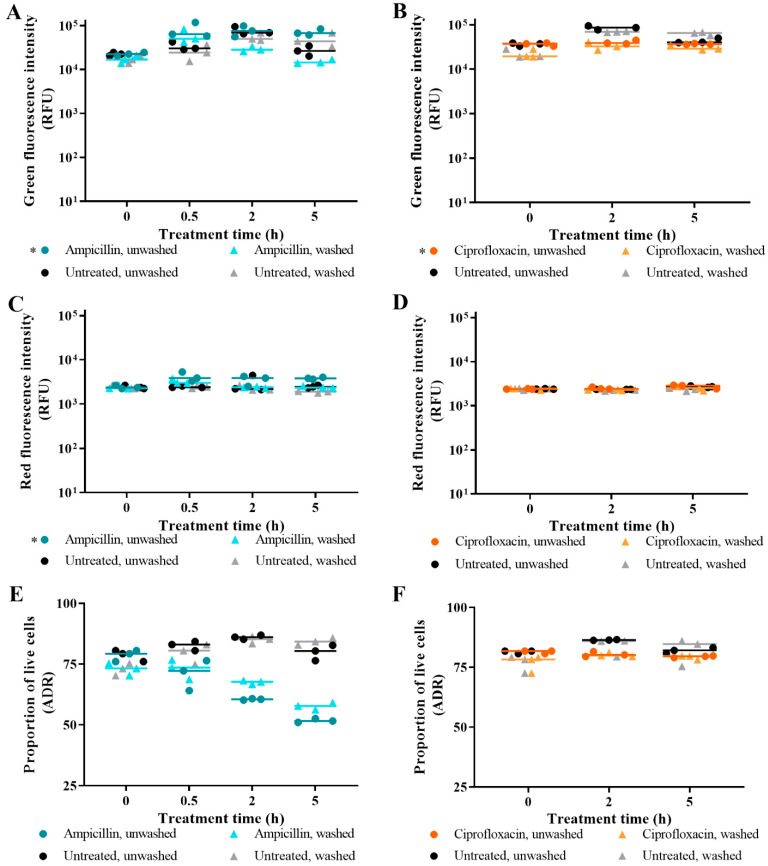
The impact of pre-staining washing of antibiotic-challenged *E. coli*. The effect of washing aliquots before staining is presented for SYTO 9-stained cells (**A**,**B**); PI-stained cells (**C**,**D**); and SYTO 9- and PI-stained cells (**E**,**F**). *E. coli* was challenged with ampicillin (teal/cyan), ciprofloxacin (orange), or nothing (untreated; black/grey), and samples were taken at 0 h (pre-treatment), 0.5 h (ampicillin only), 2 h, and 5 h. Before staining of duplicate aliquots with SYTO 9 and/or PI, one aliquot was left unwashed while the other aliquot was washed in fresh MM. Fluorescence intensities were obtained from integrating 505–515 nm for green emissions and 600–610 nm for red emissions. The adjusted dye ratio (ADR) was calculated using the green and red emissions from SYTO 9- and PI-stained samples to indicate the proportion of live cells in the aliquot. Each dot represents a biological replicate, and the median is shown by the line. Statistical significance between washed and unwashed samples is represented by * (two-way RM ANOVA, *p* value < 0.05, Tukey multiple comparisons test).

**Figure 2 microorganisms-09-00924-f002:**
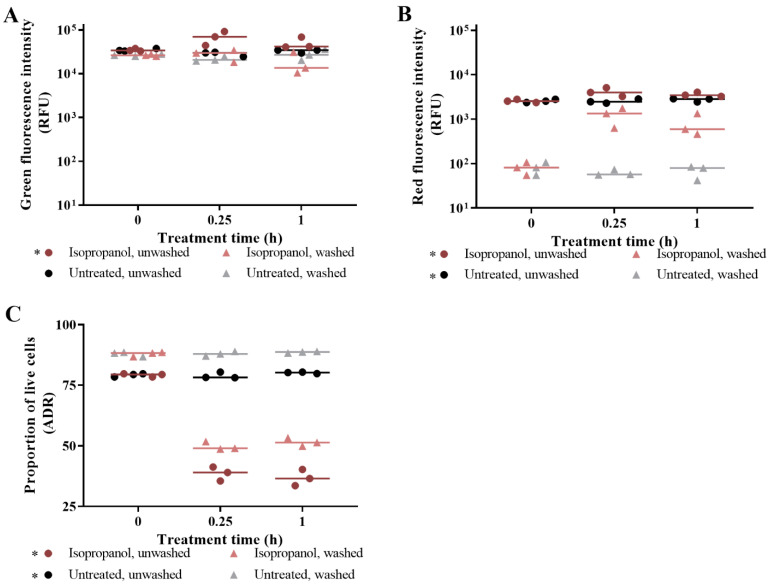
The impact of post-staining washing of isopropanol-challenged *E. coli*. The effect of washing aliquots after staining is presented for SYTO 9-stained cells (**A**); PI-stained cells (**B**); and SYTO 9- and PI-stained cells (**C**). *E. coli* was challenged with isopropanol (brown) or nothing (untreated; black/grey), and samples were taken at 0 h (pre-treatment), 0.25 h, and 1 h. Following staining of duplicate aliquots with SYTO 9 and/or PI, and one aliquot was left unwashed and the other aliquot was washed in fresh MM. Fluorescence intensities were obtained from integrating at 505–515 nm for green emissions and 600–610 nm for red emissions. The adjusted dye ratio (ADR) was calculated using the green and red emissions from SYTO 9- and PI-stained samples to indicate the proportion of live cells in the aliquot. Each dot represents a biological replicate, and the median is shown by the line. Statistical significance between washed and unwashed samples is represented by * (two-way RM ANOVA, *p* value: <0.05, Tukey multiple comparisons test).

**Figure 3 microorganisms-09-00924-f003:**
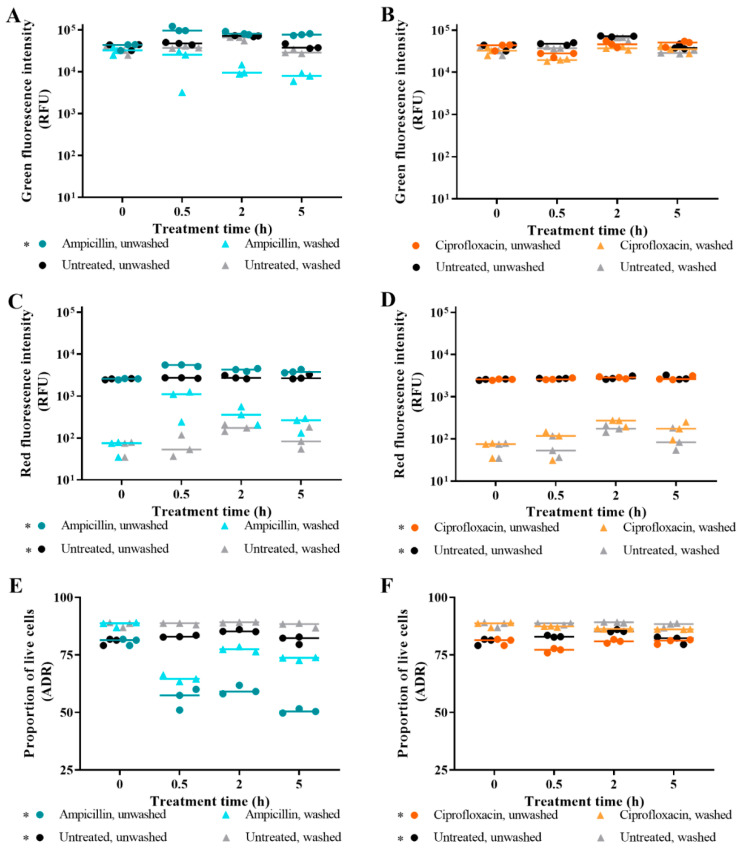
The impact of post-staining washing of antibiotic-challenged *E. coli*. The effect of washing aliquots after staining is presented for SYTO 9-stained cells (**A**,**B**); PI-stained cells (**C**,**D**); and SYTO 9- and PI-stained cells (**E**,**F**). *E. coli* was challenged with ampicillin (teal/cyan), ciprofloxacin (orange), or nothing (untreated; black/grey), and samples were taken at 0 h (pre-treatment), 0.5 h, 2 h, and 5 h. Following staining of duplicate aliquots with SYTO 9 and/or PI, and one aliquot was left unwashed and the other aliquot was washed in fresh MM. Fluorescence intensities were obtained from integrating at 505–515 nm for green emissions and 600–610 nm for red emissions. The adjusted dye ratio (ADR) was calculated using the green and red emissions from SYTO 9- and PI-stained samples to indicate the proportion of live cells in the aliquot. Each dot represents a biological replicate, and the median is shown by the line. Statistical significance between washed and unwashed samples is represented by * (two-way RM ANOVA, *p* value: <0.05, Tukey multiple comparisons test).

**Figure 4 microorganisms-09-00924-f004:**
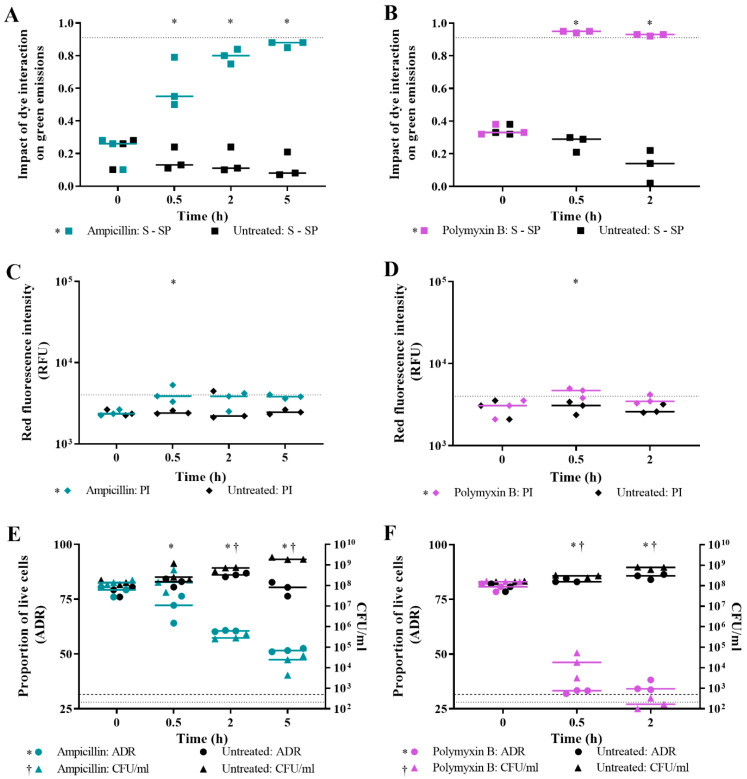
Live/dead spectrometry and culture-based enumeration of *E. coli* challenged with lytic antibiotics. At 0 h, 0.5 h, 2 h, and 5 h time points, viability of *E. coli* MG1655 challenged with ~1× MBC of ampicillin (teal), polymyxin B (purple), and an untreated control (black) was determined by measuring fluorescence of SYTO 9- and/or PI-stained cultures using the Optrode (**A**–**F**) and by culture-based enumeration (**E**,**F**). Fluorescence intensities were obtained from integrating 505–515 nm for green emissions and 600–610 nm for red emissions. The proportion of live cells in the sample population was determined by the impact of the quenching/enhancement dye interaction on green emissions (**A**,**B**); level of red emissions (**C**,**D**); and the adjusted dye ratio (ADR; **E**,**F**). The dotted lines represent the equivalent value for isopropanol-killed cells. The limit of detection for culture-based enumeration is 500 CFU/mL (dashed line). Data presented is from three biological replicates with a line plotted at the median. Statistical significance for the difference between treated and untreated at each time point is represented by * for fluorescence and † for plate counts (two-way RM ANOVA, *p* value: <0.05, Sidak’s multiple comparisons test).

**Figure 5 microorganisms-09-00924-f005:**
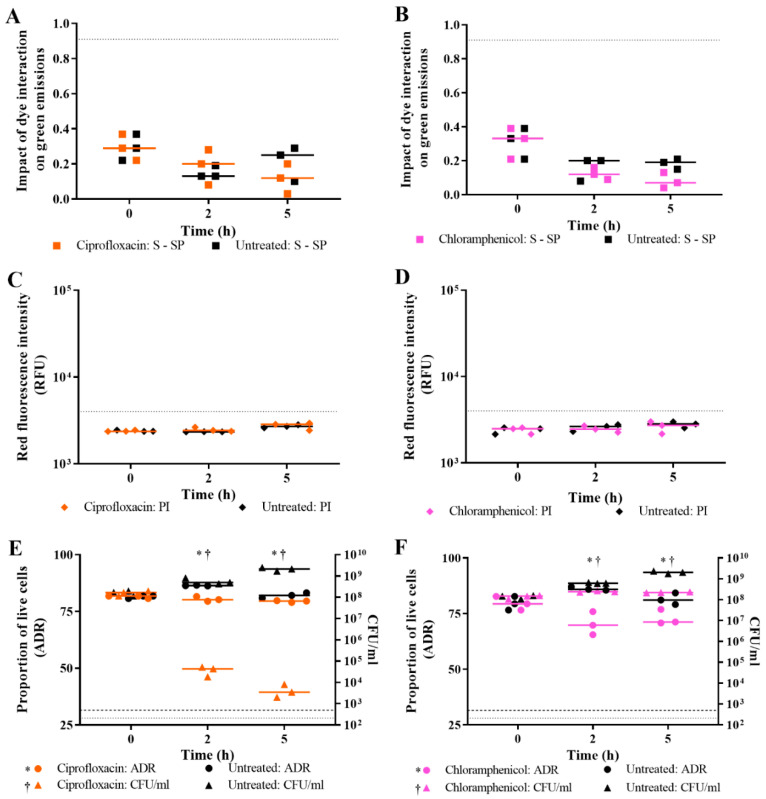
Live/dead spectrometry and culture-based enumeration of *E. coli* challenged with non-lytic antibiotics. At 0 h, 2 h, and 5 h time points, viability of *E. coli* MG1655 challenged with ~1× MBC of ciprofloxacin (orange), ~1× MIC of chloramphenicol (pink), and an untreated control (black) was determined by measuring fluorescence of SYTO 9- and/or PI-stained cultures using the Optrode (**A**–**F**) and by culture-based enumeration (**E**,**F**). Fluorescence intensities were obtained from integrating 505–515 nm for green emissions and 600–610 nm for red emissions. The proportion of live cells in the sample population was determined by the impact of the quenching/enhancement dye interaction on green emissions (**A**,**B**); level of red emissions (**C**,**D**); and the adjusted dye ratio (ADR; **E**,**F**). The dotted lines represent the equivalent value for isopropanol-killed cells. The limit of detection for culture-based enumeration is 500 CFU/mL (dashed line). Data presented is from three biological replicates with a line plotted at the median. Statistical significance for the difference between treated and untreated at each time point is represented by * for fluorescence and † for plate counts (two-way RM ANOVA, *p* value: <0.05, Sidak’s multiple comparisons test).

**Figure 6 microorganisms-09-00924-f006:**
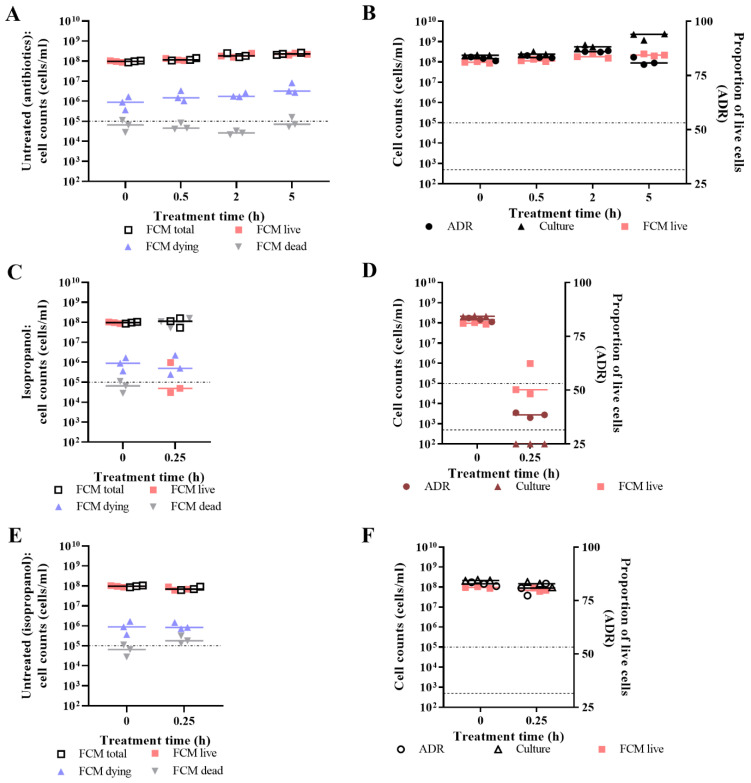
Flow cytometry of untreated *E. coli* and *E. coli* challenged with isopropanol. At 0 h and 0.25 h time points, viability of *E. coli* MG1655 challenged with 70% isopropanol (**C**,**D**) or nothing (untreated) at 37 °C (**A**,**B**) and 28 °C (**E**,**F**) was determined by measuring fluorescence of SYTO 9- and PI-stained cultures using the Optrode (**B**,**D**,**F**) and a LSR-II flow cytometer (**A**,**C**,**E**) and by culture-based enumeration (**B**,**D**,**F**). Measurements of untreated cells at 37 °C were taken at 0 h, 0.5 h, 2 h, and 5 h time points to reflect the time points for antibiotic challenge, while for untreated cells at 28 °C, measurements were taken at 0 h and 0.25 h to reflect the isopropanol challenge. FCM generated a total cell count, live cell count, dying cell count, and dead cell count (**A**,**C**,**E**). The dot-dash line represents the limit of quantification for FCM (1 × 10^5^ cells/mL). For Optrode measurements, fluorescence intensities were obtained from integrating 505–515 nm for green emissions and 600–610 nm for red emissions. The proportion of live cells in the sample population was determined by the adjusted dye ratio (ADR; **B**,**D**,**F**). The limit of detection for culture-based enumeration is 500 CFU/mL (dashed line). Data presented is from three biological replicates with a line plotted at the median.

**Figure 7 microorganisms-09-00924-f007:**
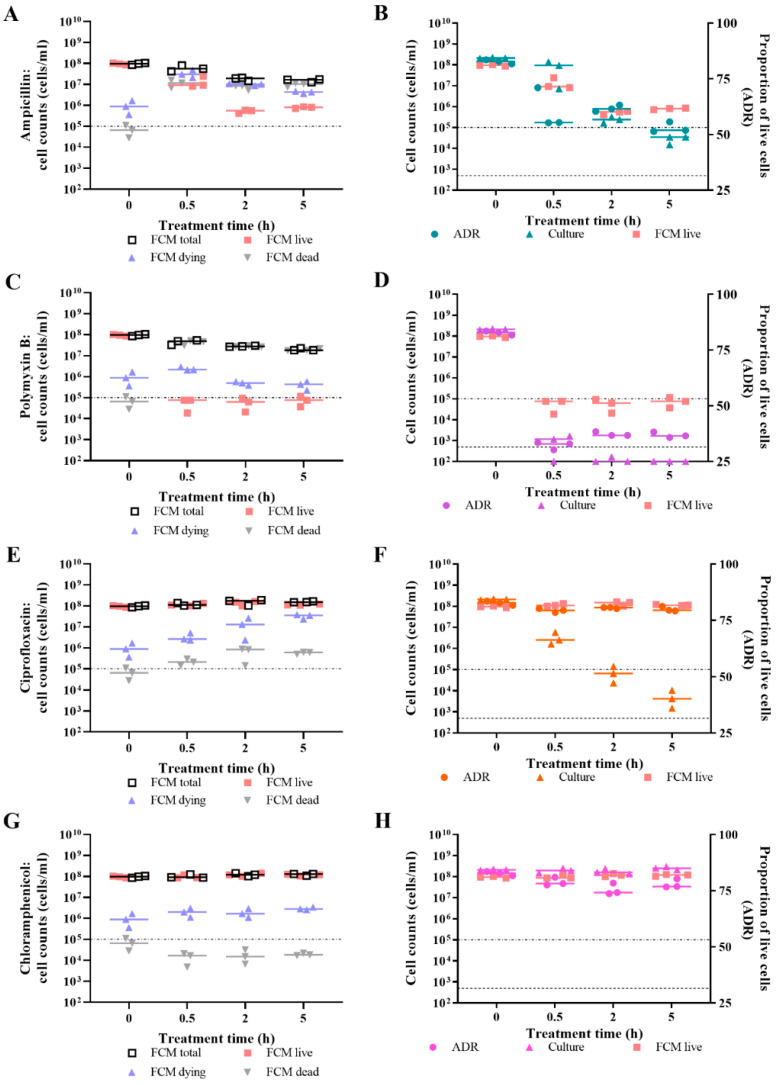
Flow cytometry of *E. coli* challenged with antibiotics. At 0 h, 0.5 h, 2 h, and 5 h time points, viability of *E. coli* MG1655 challenged with ~1× MBC of ampicillin (**A**,**B**), ~1× MBC of polymyxin B (**C**,**D**), ~1× MBC of ciprofloxacin (**E**,**F**), and ~1× MIC of chloramphenicol (**G**,**H**) was determined by measuring fluorescence of SYTO 9- and PI-stained cultures using the Optrode (**B**,**D**,**F**,**H**) and a LSR-II flow cytometer (**A**,**C**,**E**,**G**) and by culture-based enumeration (**B**,**D**,**F**,**H**). FCM generated a total cell count, live cell count, dying cell count, and dead cell count (**A**,**C**,**E**,**G**). The dot-dash line represents the limit of quantification for FCM (1 × 10^5^ cells/mL). For Optrode measurements, fluorescence intensities were obtained from integrating 505–515 nm for green emissions and 600–610 nm for red emissions. The proportion of live cells in the sample population was determined by the adjusted dye ratio (ADR; **B**,**D**,**F**,**H**). The limit of detection for culture-based enumeration is 500 CFU/mL (dashed line). Data presented is from three biological replicates with a line plotted at the median.

**Figure 8 microorganisms-09-00924-f008:**
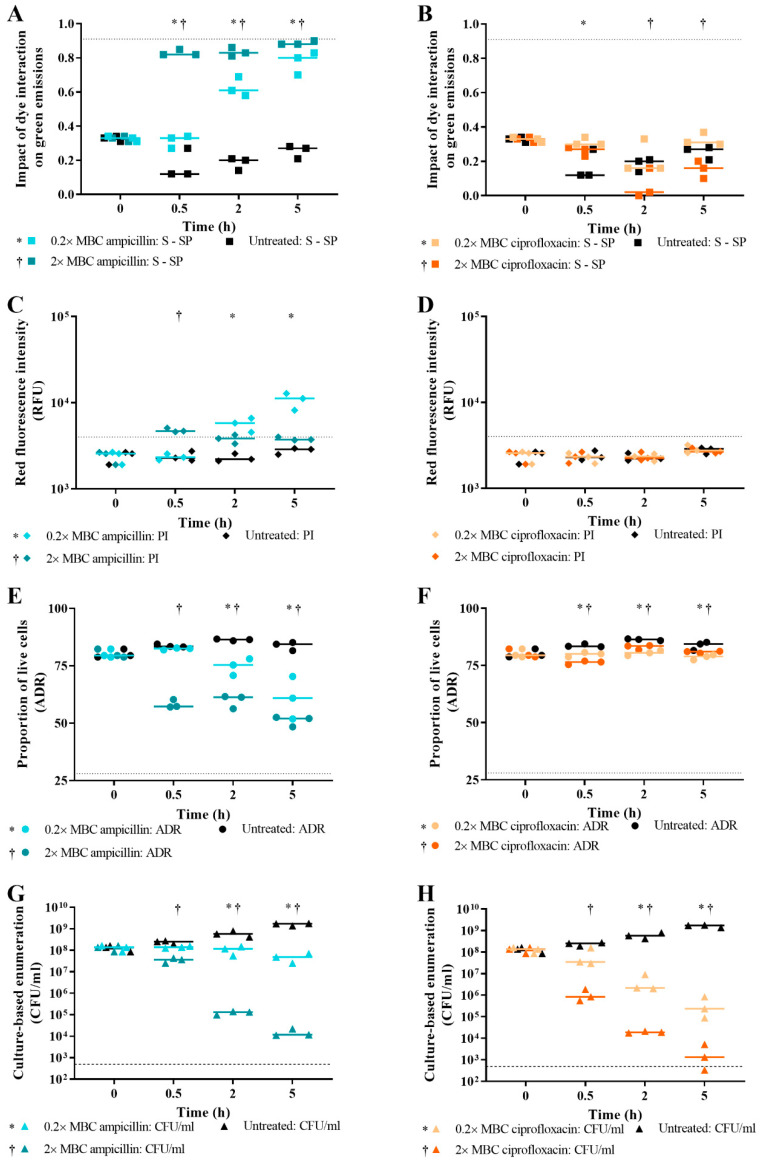
The impact of antibiotic concentration on *E. coli* live/dead spectrometry. 0 h, 0.5 h, 2 h, and 5 h time points, viability of *E. coli* MG1655 challenged with ~0.2× and 2× MBC of ampicillin (cyan and teal, respectively) and of ciprofloxacin (light orange and dark orange, respectively) and an untreated control (black) was determined by measuring fluorescence of SYTO 9-, PI-, and SYTO 9- and PI-stained cultures using the Optrode and by culture-based enumeration (**G**,**H**). Fluorescence intensities were obtained from integrating 505–515 nm for green emissions and 600–610 nm for red emissions. The proportion of live cells in the sample population was determined by the impact of the quenching/enhancement dye interaction on green emissions (**A**,**B**); level of red emissions (**C**,**D**); and the adjusted dye ratio (ADR; **E**,**F**). The dotted lines represent the equivalent value for isopropanol-killed cells. The limit of detection for viable cell plate counts is 500 CFU/mL (dashed line). Data presented is from three biological replicates with a line plotted at the median. Statistical significance for the difference between ~0.2× MBC treated and untreated is represented by * and between ~2× MBC treated and untreated is represented by † (two-way RM ANOVA, *p* value: <0.05, Tukey multiple comparisons test).

**Figure 9 microorganisms-09-00924-f009:**
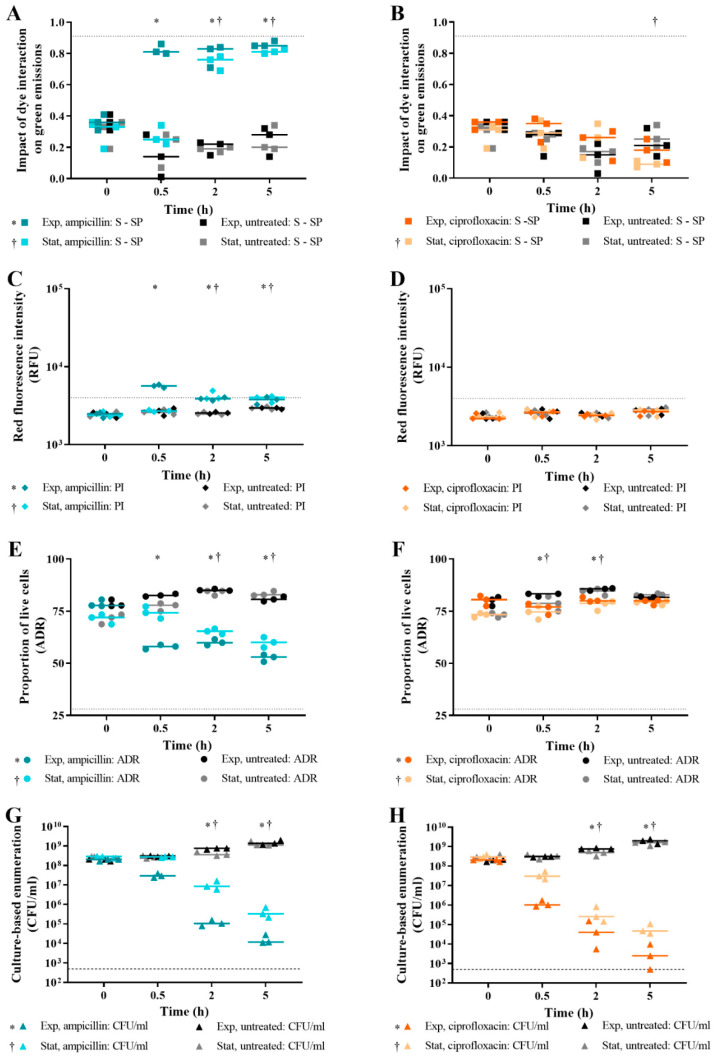
The impact of growth phase on *E. coli* live/dead spectrometry. At 0 h, 0.5 h, 2 h, and 5 h time points, viability of exponential- (exp) and stationary- (stat) phase *E. coli* MG1655 challenged with ~1× MBC of ampicillin (teal and cyan, respectively) and of ciprofloxacin (dark orange and light orange, respectively) and an untreated control (black) was determined by measuring fluorescence of SYTO 9-, PI-, and SYTO 9- and PI-stained cultures using the Optrode and by culture-based enumeration (**G**,**H**). Fluorescence intensities were obtained from integrating 505–515 nm for green emissions and 600–610 nm for red emissions. The proportion of live cells in the sample population was determined by the impact of the quenching/enhancement dye interaction on green emissions (**A**,**B**); level of red emissions (**C**,**D**); and the adjusted dye ratio (ADR; **E**,**F**). The dotted lines represent the equivalent value for isopropanol-killed cells. The limit of detection for viable cell plate counts is 500 CFU/mL (dashed line). Data presented is from three biological replicates with a line plotted at the median. Statistical significance for the difference between exponential phase treated and untreated is represented by * and between stationary phase treated and untreated is represented by † (two-way RM ANOVA, *p* value: <0.05, Tukey multiple comparisons test).

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
