# Peer review of "Rapid Detection of Escherichia coli Antibiotic Susceptibility Using Live/Dead Spectrometry for Lytic Agents"

_microorganisms, 2021, doi:10.3390/microorganisms9050924_

Round 1

Reviewer 1 Report

GENERAL COMMENTS

Although I have been critical with my comments, I like the work. I have several general comments that may improve the manuscript.

First, it is too long, and there is too much redundancy. Readers are in a hurry, and your paper will have more impact if you shorten it considerably. You may be able to shift supporting figure panels to supplementary material and shorten paragraphs. Make it as easy as possible for the reader.

Second, you did not acknowledge colleagues for critical comments. Getting local reviewers to help put together a concise story is very important, especially if you can find ones who are familiar with antibiotics. That will save you from making statements that suggest unfamiliarity with the field.

Third, I would focus on the positive statements that you can make.  It seems to me that one of those is that if you detect killing, it is likely that the cells are susceptible. Detecting resistance by your method is not yet feasible due to getting clinical isolates to grow.

Fourth, readers may pick up English-usage errors. Overall, the English is excellent, but you have a problem with compound adjectives requiring hyphens.

On the positive side, the figures were easy to understand: I did not have to look at the legends.

Below are a few specific comments that I hope you find useful.

SPECIFIC COMMENTS

Line 3 I would suggest adding a phrase “for lytic agents” to the title

Line 25 viability: this is untrue. Susceptibility measures inhibition of growth, not viability. This small error suggests the authors are unfamiliar with the antibiotic literature and seriously undermines their credibility.

I would suggest that you carefully define the AST tests early in the introduction. The key idea for the present work is that blocking growth, which is measured by MIC and reflects resistance, is upstream from the form of death that derives from a metabolic shift and accumulation of ROS. An increase in MIC will necessarily allow less killing even though the two are mechanistically distinct. A loss of killing does not always mean a change in MIC. When this occurs, it is called tolerance. On top of these ideas are killing mechanisms in which direct physical damage occurs, such as cell lysis and chromosome fragmentation. Your assay of cell lysis is a subset of killing. You need to check to see whether loss of lysis is associated with a change in MIC, the standard measure of AST.

You might find two reviews useful, one by Balaban (Nat Rev Microbiol 2019) and one by Drlica (Exp Rev Antiinfectives 2020). Although these reviews are recent, the ideas are very old, which makes it even more important to describe them carefully.

Line 65 please explain ADR so the reader does not have to refer to the original work to understand.

Line 89 bactericidal and antibiotic.  I suggest that you be more precise here to show clarity of thought: cell lysis and antibiotics that lyse cells. 

What is the role of cipro and Cam in your experiments?  Are these controls for lysis?

Line 129 what is the rationale for using MBC rather than fold MIC, since MIC is the AST standard?

Line 131 knockdown is imprecise

Line 136 slower action of cipro?  Cipro inhibits DNA replication so fast that you cannot measure it with normal microbiological methods (less than a minute)? Moreover, inhibition of DNA synthesis correlates with MIC, the standard for AST.

Line 279 as above, improve precision by saying lysis-based antibiotic-mediated killing. Also note that compound adjectives require a hyphen (this error occurs throughout the manuscript).

Line 590 death has not occurred. This seems to be backwards. Your goal is to make the reading and understanding rapid, so avoid making the reader think.  Here you know that cipro kills, but your assay, as expected, does not report that. 

Line 608 opposite. Meaning is unclear. State the result.

Line 611 at low concentration, perhaps you have repair during the long incubation on plates.

Line 684 reader now expects the Discussion to explain the effects of conc for amp, since there is an old dogma that b-lactams are time-dependent killers. Note that not all killing by b-lactams is via cell lysis: some by ROS.

Line 692. This is not a complex paper: most readers can get the message by simply looking at the figures. They will then look at the first paragraph of the Discussion. For them ADR is an undefined acronym, a big stumbling block.

Line 733. This is not completely true, because the standard AST uses inhibition of growth, not killing.  To measure inhibition of growth, you need to have growing cells.  Thus, this statement undermines author credibility. As an aside, the general view in the clinical field, is that blocking growth is sufficient for cure. I think a reference is Wald-Dickler. Thus, measuring killing is the wrong assay and your paper needs to be re-written with a sharper focus on what you have really shown.

Line 756. I think this means that a practical assay must make sure the clinical specimen is growing. This seems to be difficult and therefore limits your work. I think that what you can say is that if you detect killing by your assay, the specimen is very likely to be susceptible. But your cannot say whether it is resistant.

Line 767. Readers will want to see a comparison of cipro and chloramphenicol. The expectation is that there would be little or no difference.

Reviewer 2 Report

The authors described in a complex way the action of three different antibiotics (with different concentrations) on E. coli, studding their effect after different incubation time periods. Overall the article is comprehensive described in sufficient details starting with procedures, but also a good interpretation of results is presented. I have some comments, questions and suggestions that need clarification before suggesting the acceptance of manuscript. Please see my comments below.

Please briefly discuss also MALDI as possibility for bacteria detection in the introduction part. Also flow cytometry and should be better discussed and referenced. Life/dead ratio was previously evaluated using these techniques here DOI: 10.1016/j.ab.2019.113407. Also, it was proved that growth media type can influence the antibiotic action on bacteria. Please include these aspects also.

Lines 107-119 – please mention the purity of substances where appropriate.

Lines 121-126 - Please mention the value of OD600 for MM

Lines 128-130 - How MIC and MBC were determined? By authors or taken from literature? I don’t really agree with the idea that generally MBC was used, but in one case MIC. I understand the explanation, but maybe it would have been better to replace chloramphenicol with another antibiotic.

Lines 134-135 the authors wrote “For polymyxin B, aliquots were taken at 0 h, 0.5 h, and 2 h; a 5 h time point was not required due to the rapid action of this antibiotic” – my question is how they knew this without testing?

Lines 135-136 – the authors wrote “For ciprofloxacin and chloramphenicol, aliquots were taken at 0 h, 2 h, and 5 h, reflecting the slower action of these antibiotics” – I have the same question, how they suppose this without sampling 0.5 h?

Supplementary material is not mentioned in the main text.

Round 2

Reviewer 1 Report

GENERAL COMMENTS

The work has improved considerably. However, it still has considerable redundancy, making the Discussion more like a Summary.  Compound adjectives still require attention.  Below are a few minor comments.

SPECIFIC COMMENTS

Line 15  however clause is awkward

Line 475 and elsewhere: you need a “thus” sentence at the end of your very long paragraphs. Also, repeating the statistical assay with every piece of data is unnecessary and distracting. 

Line 763. The singular of media is medium

Line 824 false. Which is really false? Your assay or plate assays with long incubation times in the absence of drug? During the long incubation repair can occur. Which reflect the clinical situation better?

Reviewer 2 Report

The authors addressed all my requirements and/or justified well their answer. I suggest the acceptance of the manuscript in the current form.

Author Response

No changes required